# Low cost DNA data storage using photolithographic synthesis and advanced information reconstruction and error correction

Philipp L. Antkowiak [1], Jory Lietard[2], Mohammad Zalbagi Darestani[3], Mark M. Somoza [2,4,5], Wendelin J. Stark[1], Reinhard Heckel[3,6 ✉] & Robert N. Grass [1 ✉]

Due to its longevity and enormous information density, DNA is an attractive medium for archival storage. The current hamstring of DNA data storage systems—both in cost and speed—is synthesis. The key idea for breaking this bottleneck pursued in this work is to move beyond the low-error and expensive synthesis employed almost exclusively in today's systems, towards cheaper, potentially faster, but high-error synthesis technologies. Here, we demonstrate a DNA storage system that relies on massively parallel light-directed synthesis, which is considerably cheaper than conventional solid-phase synthesis. However, this technology has a high sequence error rate when optimized for speed. We demonstrate that even in this high-error regime, reliable storage of information is possible, by developing a pipeline of algorithms for encoding and reconstruction of the information. In our experiments, we store a file containing sheet music of Mozart, and show perfect data recovery from low synthesis fidelity DNA.

[1] Department of Chemistry and Applied Biosciences, ETH Zürich, Vladimir-Prelog-Weg 1-5, 8093 Zürich, Switzerland. [2] Institute of Inorganic Chemistry, Faculty of Chemistry, University of Vienna, Althanstraße 14, A-1090 Vienna, Austria. [3] Department of Electrical and Computer Engineering, Rice University, 6100 Main St., Houston, TX 77005, USA. [4] Chair of Food Chemistry and Molecular Sensory Science, Technical University of Munich, Lise-Meitner-Straße 34, 85354 Freising, Germany. [5] Leibniz-Institute for Food Systems Biology at the Technical University of Munich, Lise-Meitner-Straße 34, 85354 Freising, Germany. [6] Department of Electrical and Computer Engineering, Technical University of Munich, Theresienstr. 90, 80333 Munich, Germany. ✉email: reinhard.heckel@gmail.com; robert.grass@chem.ethz.ch

A few years ago, DNA was (re-)introduced as a powerful tool to store digital information in chemically synthesized molecules: Goldman et al. and Church et al. nearly simultaneously presented methods to store about 1 MB of digital information in DNA[1,2]. The rise of DNA as a potential storage medium has been fueled by recent progress in parallelized DNA synthesis and DNA sequencing technologies, such as portable sequencing devices (Oxford Nanopore MiniOn and Illumina iSeq 100). Moreover, recent enzymatic methods have been considered for the sequence-controlled synthesis of DNA[3,4]. While an interesting alternative to the traditional phosphoramidite methods for DNA synthesis, enzymatic methods are still a step away from generating highly variable DNA libraries.

In this paper, we pursue the idea of employing a traditional chemical synthesis strategy but using lower-cost units operating at higher speeds and lower precision. While this is theoretically possible, it results in significantly higher sequence error levels. Even though high error levels are prohibitive for most biological applications of long DNA oligos, a DNA data storage system can, in principle, cope with very high error rates and enable reconstruction of the information from noisy reads through the use of advanced error correction algorithms.

Here we show that it is possible to store data in very noisy photochemically synthesized DNA by encoding nearly 100 kB of digital information into DNA sequences, synthesized economically using a rapid light-directed array platform. To enable storage, we present data handling routines capable of perfectly recovering the information from the DNA, even when the synthesized molecules contain far more synthesis errors than has been the case for any preceding digital DNA storage work.

## Results

**DNA synthesis and design strategy**. In this work, we explore light-directed maskless array technology for DNA synthesis. In contrast to the electrode array-based technology commercialized by CustomArray and the material deposition technologies (printing) utilized by Agilent and Twist Biosciences, light-directed synthesis promises greater scalability at lower cost. Additionally, the technological basis of maskless array synthesis, i.e., digital micromirror devices (DMD), is highly accessible having most recently revolutionized 3D printing and pocket sized image projectors[5,6]. Light-directed DNA synthesis utilizing chrome photolithography masks have a tradition in the commercial production of DNA microarrays[7]. The successful replacement of the masks with digital micromirrors was described by Singh-Gasson et al. in 1999[8]. This maskless approach is essential for digital DNA storage as it enables fast synthesis of arbitrary sequences with minimal hardware. Our set-up follows the original design of Singh-Gasson et al. and consists of a flow cell, on to which UV light patterns are directed via a digital micromirror device[8,9]. The flow cell is supplied in a cyclic manner by an oligonucleotide synthesizer loaded with standard solvents and reagents in addition to phosphoramidites with photolabile 5′ hydroxyl protecting groups (Bz-NPPOC) and a photo-exposure reagent.

Since the coupling yields for this technology are relatively low (95–99% chemical yield)[8,10], we decided to aim for the synthesis of 60 nt long oligos, which are significantly shorter than previously utilized oligos for DNA storage applications (117–159 nt)[1,2,11]. As a result, we refrained from synthesizing sequence amplification sites (usually ca. $2 \times 20$ nt)[1,2,11–13] at the two ends of the oligos. For short sequences this overhead nearly exceeds the synthesis effort for the actual information storage and is an inefficient use of resources. Consequently, we could utilize the full 60 nt length for the storage of information but had to find

means to generate dsDNA sequence libraries from the short ssDNA oligos in down-stream processing steps (see further below). In addition to that, we decided not to use the full resolution of the DMD device ($1024 \times 768$ pixels), but to illuminate spots consisting of $8 \times 6$ pixels together, resulting in a final exposure resolution of $128 \times 128 = 16'384$ possible oligos.

**Encoding 100 kB of data into DNA**. As every nucleotide can at most store 2 bits of information, 16,383 oligos each 60 nt long have a storage capacity with an upper bound of 245,745 bytes. Based on our previous work on DNA data storage, and the higher expected error rates of the array-based synthesis methods, we decided to use a conservative 60% of the nucleotides for error correction code redundancy, leaving the possibility of storing ca. 100 kB of digital information. An overall scheme of the data storage channel can be found in Supplementary Fig. 5. We chose to store 52 pages of digitalized sheet music, in detail the String Quartet "The Hunt" K458 by Wolfgang Amadeus Mozart, digitized by Project Gutenberg and publicly available as four MusicXML files (3.6 MB). The files were compressed (OS X, zip) yielding 99,103 bytes of digital information.

The encoder and decoder were jointly designed taking the expected errors introduced by the light-directed synthesis into account. While many previous DNA storage encoding approaches included measures to exclude specific DNA patterns (homopolymers, GC content), we deliberately did not include such measures into our approach. Instead, our approach starts with the invertible pseudo randomization of the data, which elegantly avoids long homopolymer stretches and oligos with highly unbalanced GC contents that become very unlikely via multiplication of pseudorandom sequences. The negligible increase in reading error rates due to rare homopolymers is more effectively dealt with using our error-correcting mechanisms, as opposed to actively avoiding them. Additionally, randomization yields sequences that are pairwise close to orthogonal to each other, which is desirable for distinguishing them, for example, by a clustering algorithm. Randomization was first used by Organick et al. to avoid unbalanced GC contents[13].

Next, the original data stream was organized into 10977 sequences of length $14 \times 6$ bits, which are seen as 6 blocks of 14 bits for our outer decoding strategy. To this end, a Reed-Solomon code of length 16,383 with symbols in $\{0, \dots, 2^{14-1}\}$ generates $16,383 - 10,977 = 5406$ new sequences containing redundant information of the original 10,977 sequences. In the next step, we add a unique index of length 24 bits to each of the 16,383 sequences. The sequences now have length $24 + 14 \cdot 6 = 18 \cdot 6$ bits. Consequently, the inner decoder can view each sequence as consisting of 18 symbols of length 6 bits, adding two parity symbols by inner decoding with a Reed-Solomon code with symbols in $\{0, \dots, 2^{6-1}\}$, yielding 16383 sequences of length $20 \cdot 6$ bits. Finally, we map each sequence to a DNA sequence of length 60 nt by using the map $00 \rightarrow A, 01 \rightarrow C, 10 \rightarrow G, 11 \rightarrow T$. In this final step, the placement of the information within the sequence is around the index so that the index is not at the beginning (or end) of the oligo, as error levels at these locations were expected to be higher (see Fig. 1). More details can be found in the Supplementary Note 1.

**Maskless array DNA synthesis performance**. In this section, we quantify the errors introduced during synthesis. Following the synthesis of the 16,383 sequences using the maskless array synthesis device described above, the oligos were cleaved from the solid support and had to be made compatible for sequencing. As the resulting ssDNA does not contain any constant sequence stretch usable as primer site, we had to find other means to

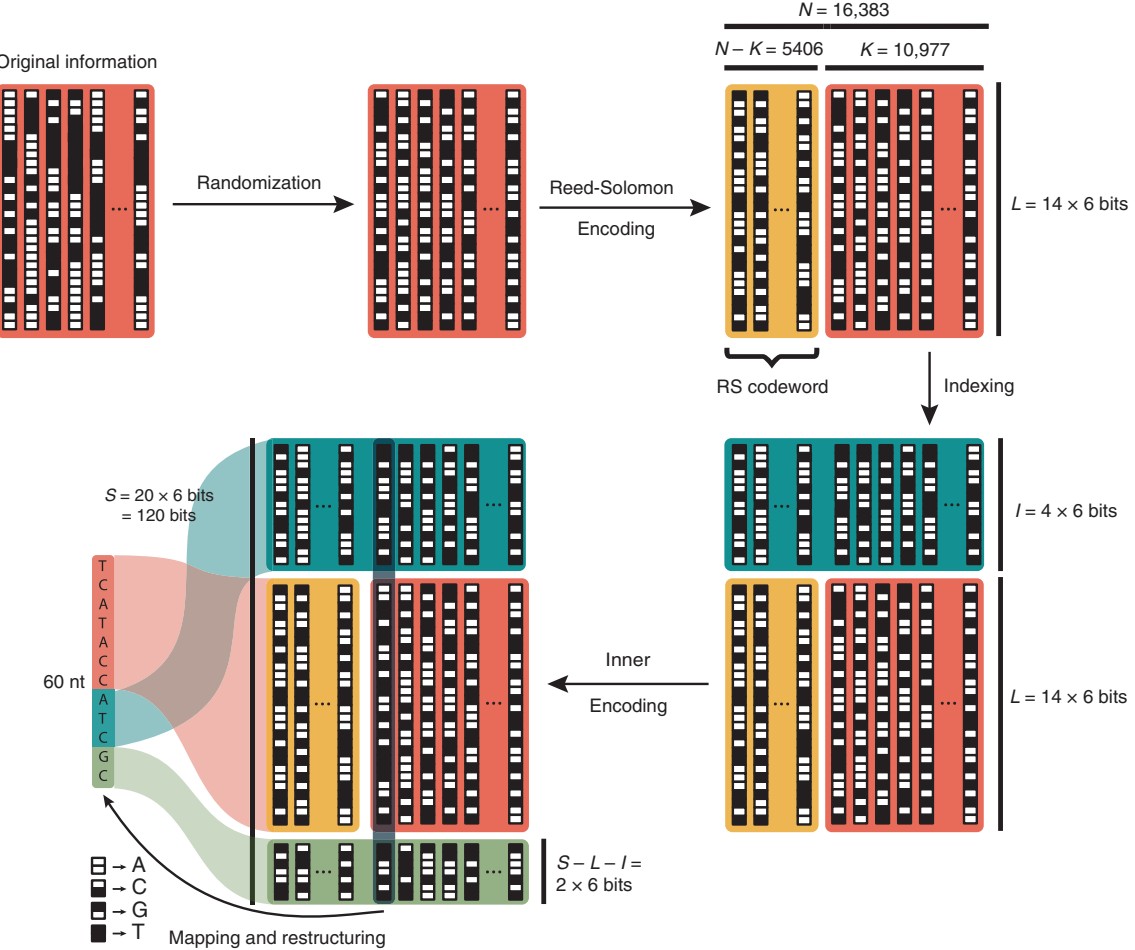

**Fig. 1 Encoding scheme for storing 100 kB of music data in 16,383 DNA sequences.** The stored information is structured as 14 packages containing 6 bits ($L$) with a length of $K = 10,977$ sequences. Via multiplication of a pseudorandom sequence, the dataset is randomized and subsequently encoded with Reed-Solomon codes. Addition of indices ($I$) and redundancy from the inner encoding step yields $N = 16,383$ sequences with a length of $S = 120$ bits, respectively 60 nt.

prepare the DNA for sequencing, which requires dsDNA with well-defined adapter sequences attached to both ends of the sequence to enable Illumina flow cell binding. Usually such adapter sequences are introduced via A-tailing and ligation to dsDNA, or overhang-PCR, but neither was possible with the ssDNA pool at hand. For the introduction of adapter sequences, we tested a ssDNA library preparation kit from Swift Biosciences, which sequentially attaches constant adapters to ssDNA fragments, thereby allowing the formation of dsDNA and the additions of sequencing adapters[14]. The underlying principle of library preparation out of a random ssDNA pool without a constant region is ligation of template-mediated addition of deoxyribonucleoside monophosphates (dNMPs) including an attenuator. The latter will control the number of attached dNMPs and additionally comprises part of an NGS adapter double-stranded sequence. The addition of bases to the 5′ end of the ssDNA is directed by terminal deoxynucleotidyl transferase (TdT) and concomitant ligation of both ends. Similarly, to the 3′ end of the oligonucleotide, a DNA polymerase adds bases complementary to a homopolymeric attenuator-adapter construct that is ligated in the same step. In this way, the treated ssDNA can undergo the full adapter ligation in a PCR that also incorporates read indices. We provide a more detailed analysis and empirical data on the kit functionality in the Supplementary Note 2.

Together with the required Illumina primers, the kit also adds a polynucleotide tail with a median length of 8 bases (80% G, 20%

A) to the 3′ end of each fragment. It is known to efficiently enable library preparation with inputs as low as 10 pg and even read coverage[14]. DNA sequencing on Illumina NextSeq yielded 30 million reads with an average length of 63 nt[15] (see Fig. 2a). The increase in reads from the specified 60 nt to 63 nt can be explained by the polynucleotide tail introduced during library preparation (in conjunction with deletion errors, see discussion further below).

An analysis of the read GC content does not show any PCR and sequencing pre-prep derived preferred sequence reading (see Fig. 2b).

Compared to the specified sequences, the read sequences contain a large amount of errors. The measured error probabilities are 2.6% for substitutions, 6.2% for deletions and 5.7% for insertions (see Fig. 2c). For a fixed 60 nt analysis window, the average deletion level should be equivalent to the substitution error level. However, at the highly erroneous 3′ end, it is difficult for the alignment code to determine if an error is an insertion or a substitution. These error levels increase drastically towards the end of the sequences, and are considerably higher than the error levels of previous attempts to store information in DNA, where substitution levels of about 0.5% are common, and insertion and deletion errors are well below 0.1%[16] (see Fig. 2d). Since the DNA of the present experiment was read with equivalent technologies as in previous DNA storage experiments, the higher error levels can in principle either be caused by the

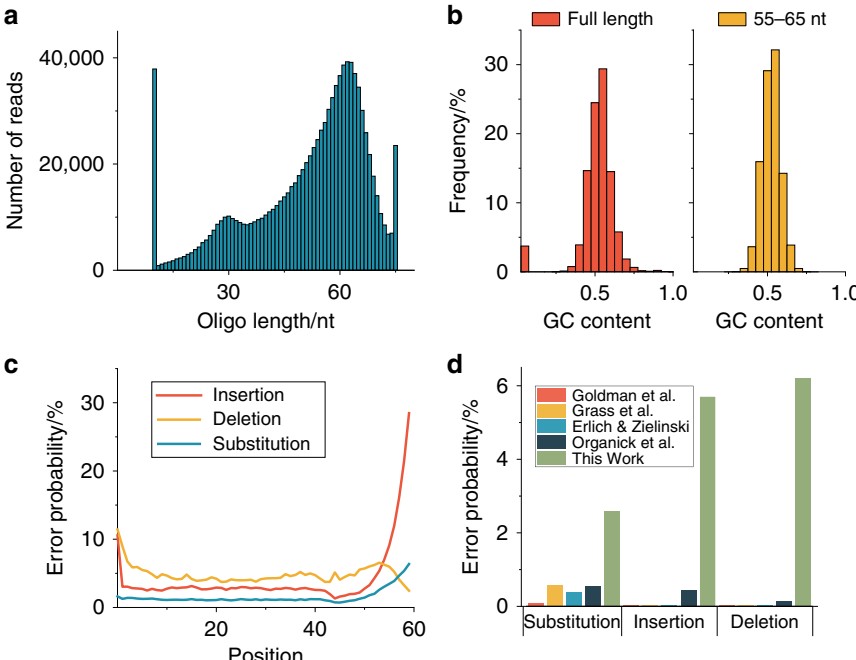

**Fig. 2 Read analysis and comparison to prior work of the photochemically synthesized DNA. a** Read length distribution ($n = 1$). **b** Normalized GC content **c** Error probability of reads cut after 60 nt in 5′–3′ direction. **d** Comparison of errors in previous work. Source data are provided as a Source data file.

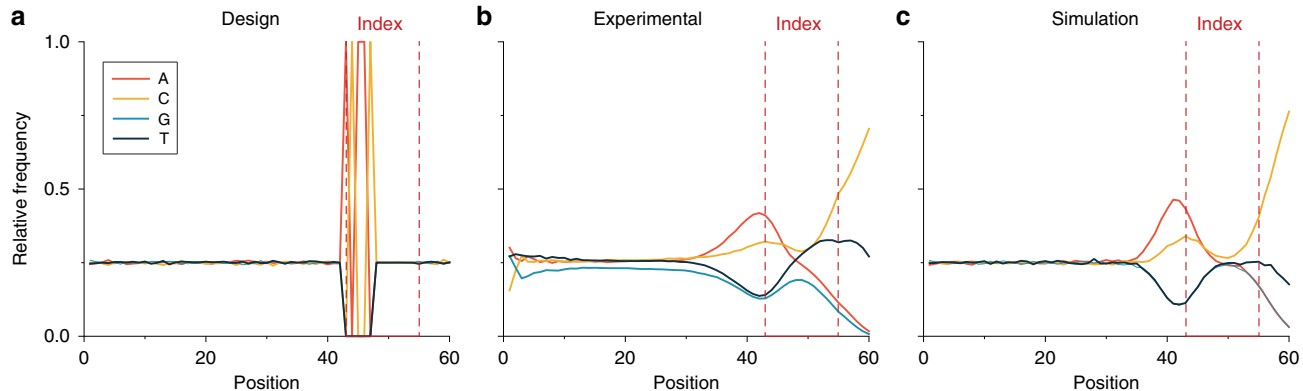

**Fig. 3 Relative frequency of DNA bases for each position. a** Relative frequency as specified by design. Note that the peaks in positions 40–50 nt are due to the index. **b** Measured relative frequencies ($n = 1$). **c** Monte-Carlo simulation with deletion error rate $\lambda = 7$ (poisson distributed probability), a substitution error rate $\lambda = 2$ and an insertion rate $\lambda = 3$. The simulated tailing ratio is C:T 5:1. Source data are provided as a Source data file.

different synthesis procedure, or the different sequencing sample preparation routine (with no primers synthesized into the sequences). As explained next, the errors can be attributed to the synthesis procedure, namely statistical nucleotide deletions prior to the index position cause the shift of the index region to the left. A way to investigate the origin of these errors is to plot the averaged relative frequencies of the individual bases along the sequences (see Fig. 3).

While there seems to be a preference for G over C for base 1 at the 5′ end of the sequence, G is slightly underrepresented for positions 2–30, with all other bases having a very similar relative frequency (see Fig. 3b). This underrepresentation of G can be explained by a lower coupling efficiency for Bz-NPPOC-dG[17]. The sudden rise of A and C close to position 40 can be explained by the position of the index (See Fig. 1). For this experiment, we chose an index size of $4 \times 6 = 24$ bits, which enables the indexing of $2^{24} = 10^7$ sequences, but only use the first 16,383 index values, meaning that from the 24 index bits, only 14 are actually used, and the first 10 bit positions (equaling 5 nt) always remain at the

default value (ACAAC). This can be seen directly from the relative frequencies of the bases of the specified sequences (Fig. 3a). A comparison between the relative frequencies of the specified sequences and the read sequences surrounding the index positions further displays that in the experimental results, the maximum for A and C bases of the index is less intense, and shifted towards the left. As previously stated, this effect can only be explained by the presence of statistical nucleotide deletions at positions prior to the index. Looking towards the end of the read sequences (Fig. 3b), a sharp rise in C and a rise in T can be observed. Both can be explained when deletion errors in earlier positions are considered: one T is appended to the 3′ end of the sequence during synthesis, and the C rich tail is added to each sequence during library preparation. This C rich tail was not expected to be visible, as it was appended at the 3′ end, over the 60 nt sequencing window. However, a high deletion rate in synthesis translates into shorter oligonucleotides, where the presence of the C rich tail[18] introduced post synthesis can now become visible during sequencing and is responsible for the high

substitution error at the 3′ end of the sequences, if looking at a fixed 60 nt information window.

In order to quantify and further prove these effects we performed a Monte-Carlo type experiment with the specified sequences described in detail in the Supplementary Methods. This experiment aims to show that a statistical introduction of deletion, insertion, and substitution errors results in the observed ACGT distribution patterns found experimentally (see Fig. 3b). Deletion, insertion and substitution errors were statistically induced and C and T rich sequences were added at 3′ positions, obtaining a desired length of 60 nt. While these Monte-Carlo computational experiments were performed with varying deletion probabilities and C rich tail sequences, the best overlap with the measured data is achieved with a deletion, substitution, and insertion probability of 11.7%, 3.3%, and 5.0 %, respectively. Although this computational analysis is fully independent of the read error analysis performed further above (Fig. 2c, comparing read sequences with specified sequences), the results on all error levels are in good agreement. The deletion errors are primarily due to deliberate photodeprotection underexposure (to reduce synthesis time and cost) and secondarily caused by coupling failures. We have previously measured the coupling efficiencies of the Bz-NPPOC monomers at 99.9% for A, C, and T, and 97.1% for G[17]. The insertion errors result from optical effects resulting in the non-perfect mapping of the digital synthesis masks to the synthesis chamber. These optical effects are due to light diffraction as well as scattering from mirror edges and from dust and imperfections of the optical elements[9]. The increased insertion error rate at the 3′ end of the sequences is a direct consequence of the synthesis deletion errors, as any sequence shorter than 60 nt is appended with the C rich tail introduced during sequence adapter attachment. Above analysis shows that the observed increase in the insertion error level at the 3′ end of the sequences is a consequence of statistical deletion errors throughout the sequence in conjunction with a fixed 60 nt analysis window. As all DNA processing steps (especially PCR and sequencing) are equivalent to previous DNA storage experiments depicting significantly lower error levels, it can be deduced that the increased statistical error levels found in this dataset can be ascribed to synthesis. To better understand how errors are introduced during synthesis, we analyzed the edit distance and number of reads corresponding to the position of the micromirror on the DMD. Figure 4 shows the spatial distribution of these characteristic numbers.

It is apparent that there is significantly higher production of DNA in a concentric manner, leaving the edges and especially the corners with a low read output (see Fig. 4b).

This can be explained by optical losses through UV LED intensity and inefficiencies in the light homogenizer, as well as uneven covering of the UV absorbing β-carotene. Less light arriving on the synthesis surface results in a lower coupling efficiency. On the other hand, the readable sequences have a more homogeneous error distribution over the entire synthesis surface. Generally, reflection and scattering effects decrease synthesis integrity[19] due to the increased mutation probability. A grid shaped pattern could stem from inefficient light homogenizing as well. Another explanation are non-random features during synthesis of index positions that have a similar grid pattern. Neighboring features that couple the same nucleotide in one step and are unintentionally photo-exposed would show a slightly reduced error rate.

It should be noted that there is a trade-off between deletion error levels and overall DNA integrity which could be improved by tuning the synthesis procedure. Longer light exposure making deletion errors less likely could, in turn, lead to photoinduced damage to the DNA. However, a goal of this work is to investigate if the data can also be recovered if the DNA is synthesized at non-optimal conditions for sequence quality by the usage of appropriate error handling schemes.

**Perfect data recovery by advanced clustering and error handling**. Due to the relatively high error levels introduced during DNA synthesis, only one in $10^7$ of the read sequences are error-free. As shown above, errors are more likely at the end of a sequence, so as a first step, we shorten the sequences longer than 60 nt to a length of 60 nt. Out of the shortened oligonucleotides, the error-free sequences are still only 0.04% of all sequences. However, one could argue that a single copy of each sequence without error is sufficient for decoding. Nevertheless, only 0.09% of the original sequences occur at least once in the 15 million sequences without error. If we again consider the sequences shortened to length 60 nt, 31% of the sequences appear at least once without error, but still too few for enabling reconstruction with the outer error correcting code alone.

Thus, the decoder is specifically designed to handle large amounts of errors. We first cluster the sequences, align the sequences in each cluster, and subsequently extract candidate sequences from the clusters. After that, we apply the inner and outer decoder to the candidate sequences. The steps are explained in detail below. Clustering of the reads: For efficient clustering of the sequences, we have developed a locality-sensitive, hashing-based clustering algorithm, described in Supplementary Note 3. We compare this approach to a naive clustering approach that simply looks at the first few (16 performed best) nucleotides of

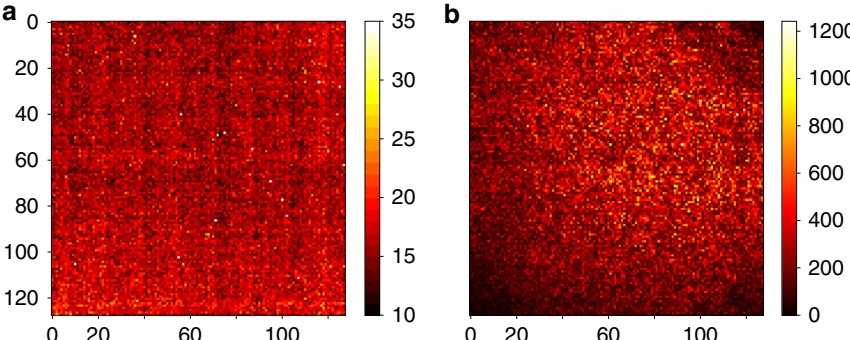

**Fig. 4 Spatial error analysis of the photo-directed synthesis machine. a** Edit distance of reads to the original sequence mapped to the synthesis position on the photolithographic chip ($n = 1$). **b** Number of reads after clustering for each position. $128 \times 128 = 16{,}384\text{-}1$ sequences were synthesized, resulting in a file size of 99.1 kB. Source data are provided as a Source data file.

each sequence. Using hash tables, the naive clustering algorithm is linear in the number of reads, and thus computationally cheap. It also produces reasonably good clusters, and enables recovery, but the locality-sensitive hashing approach gives better clusters at a slight increase in computational cost, overshadowed by the computationally most expensive step of multiple alignment. Multiple alignment and extraction of candidates: We discard all clusters with fewer than five sequences, since we found that such small clusters do not enable generations of good candidates at the error levels we observe. We then compute a multiple alignment of the clusters with 5–15 fragments using the MUSCLE[20] multiple alignment algorithm. From each cluster with more than 15 fragments, we select five randomly-chosen sets of 15 fragments, and again compute a multiple alignment. This is reasonable, since computing multiple alignments becomes very expensive in the number of sequences and did not perform better beyond the length of 15. For each alignment, we compute a candidate sequence using weighted majority voting, where the weights take into account that deletions are much more likely than substitutions and insertion errors. Specifically, we carry out majority voting of a multiple alignment by weighting the numbers of A, C, G, T with 1 and a deletion with 0.4 (See Supplementary Fig. 4) as we found those weights to give high performance. These steps are visualized in Fig. 5.

Inner decoding: We then map the candidate sequences to sequences of $6 \cdot 20$ bits, inner-decode the candidate sequence which yields sequences of length $6 \cdot 18$ bits. The sequences are then ordered according to the indices, which yield a subset of the $M = 16,383$ (ordered) sequences of length $14 \cdot 6$ each. Note that this typically only yields a subset of the sequences because a significant fraction of the sequences cannot be recovered with the previous steps. Outer decoding: finally, we outer-decode each of the outer code words. This yields $K = 10,977$ sequences of length $14 \cdot 6$, which we regard as a sequence of $14 \cdot 6 \cdot 10,977$ bits, corresponding to the reconstructed information. Note that the outer code can recover the lost sequences, provided there are not too many (specifically, the outer code can recover all sequences if no more than $1 - K/N = 33\%$ are missing).

We next discuss the results[15] we obtained by applying the coding scheme described above to the data from our experiment —the scheme did perfectly recover the data. In more detail, after the clustering and majority voting steps described above, we have 393,004 candidate sequences, out of which 7% have no error, and 74% of the original sequences appear at least once without error (for the naive clustering approach; with the locality-sensitive-hashing this number becomes 83%). Those sequences are passed to the inner decoder and then to the outer decoder. The outer decoding step reconstructs the data by correcting 3.4% erasures and 13.3% errors, and recovers the information perfectly. The erasures are sequences after the inner decoding step for which we

have no candidates, and the errors are candidates that are erroneous candidates.

We would like to emphasize that recovery without the clustering, alignment, and majority voting steps would be impossible, since, as mentioned before, only a fraction of $10^{-7}$ of the read sequences are error-free, and the errors are mostly due to imperfections in synthesis. The computationally most expensive part of the recovery pipeline is the multiple alignment step. We rely on a relatively high-quality multiple alignment and found that this step, along with the weighted majority voting is critical to generate sufficiently good candidates for the error-correcting scheme.

**Scaling to larger files allows for a substantial cost reduction**. So far, we focused on a file of about 100 kB stored on 16,383 sequences. In order to understand how our method scales to larger amounts of data, we performed two additional experiments where we stored about 323 kB and 1.3 MB on 49,149 and on 196,596 sequences, respectively, with exactly the same coding scheme and with the same parameters (in particular the same amount of redundancy) as described before. We refer to those files as file 2 and file 3, respectively, and to the original 100 kB file as file 1. For file 2 and 3 we obtained 100,706,815 and 195,619,515 many reads, respectively[15].

Compared to file 1, the error probabilities in the sequences are higher, specifically, we found the insertion probabilities to increase by about 2% and 4% for files 2 and 3 relative to file 1, whereas the substitution and deletion errors remained relatively constant (see Fig. 6, and compare to Fig. 2 panel c). Note that we expect the error probability to be higher since when synthesizing a larger amount of sequences, the spots on the array become smaller, and thus we expect lower-quality synthesis.

Due to the higher amount of errors, we were not able to recover the information, but would have been able by using more redundancy in our coding scheme. In more detail, after clustering the reads of file 2 and file 3, 35% and 31% of the original sequences appear at least once without error, as opposed to 83% for file 1. By increasing the redundancy of the outer code by a factor of about 3, we would be able to recover the data perfectly from those measurements. In this case, costs would be further minimized to 160 US\$ MB$^{-1}$ for file 2 and 40 US\$ MB$^{-1}$ for file 3.

## Discussion

Previous approaches to DNA storage have used high quality DNA strands supplied from commercial suppliers (Twist, Agilent and CustomArray). While Twist and Agilent use material deposition (printing) methods for the formation of individual oligos, CustomArray uses electrochemistry for site-specific deprotection (see Fig. 7, more details available in Supplementary Note 4). As shown

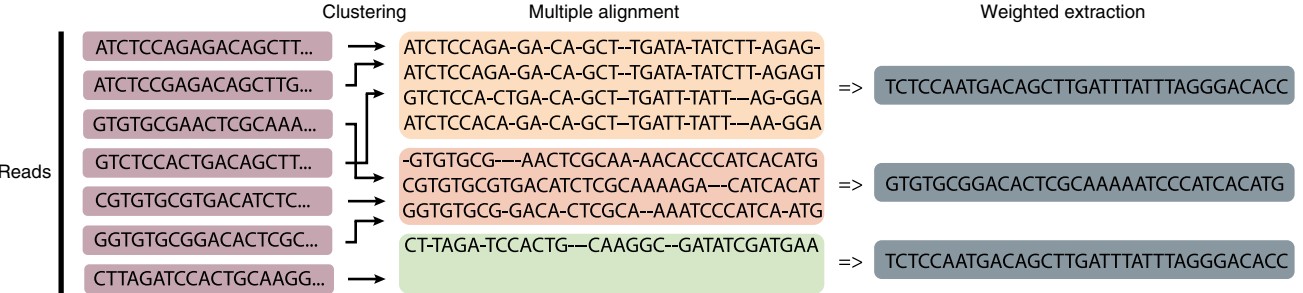

**Fig. 5 Clustering and extraction of read DNA sequences.** The sequenced DNA fragments are clustered, aligned and possible candidates are extracted in a majority voting step. The alignment is visually enhanced by marking deletion errors with a dash.

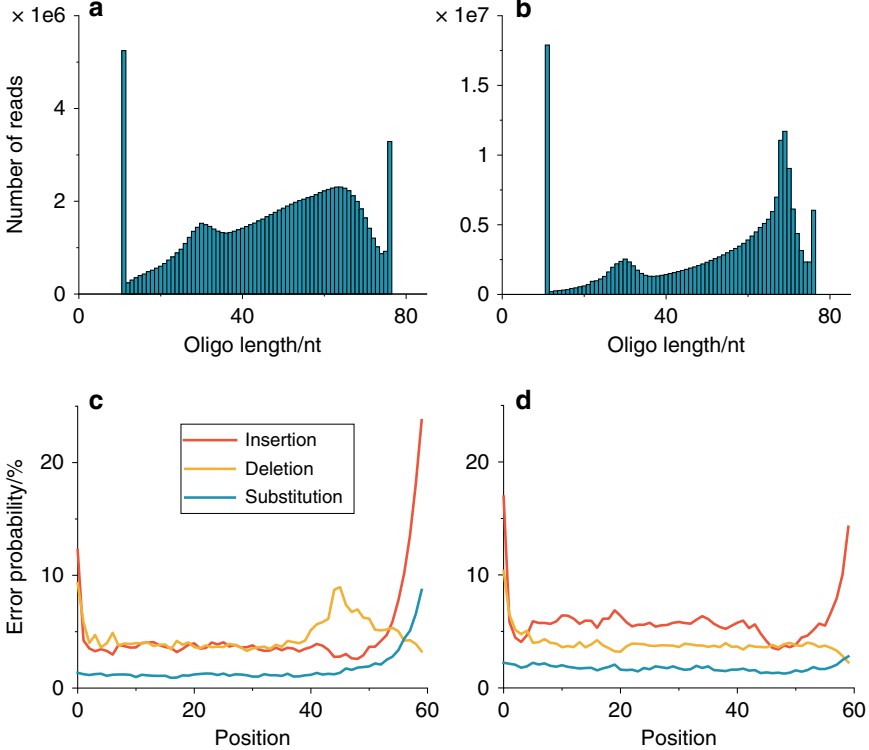

**Fig. 6 Read analysis for files of larger size. a, b** Read length distribution for file 2 ($n = 1$) and file 3 ($n = 1$), respectively. **c, d** Error probability of reads cut after 60 nt in 5′–3′ direction for file 2 and file 3, respectively. Source data are provided as a Source data file.

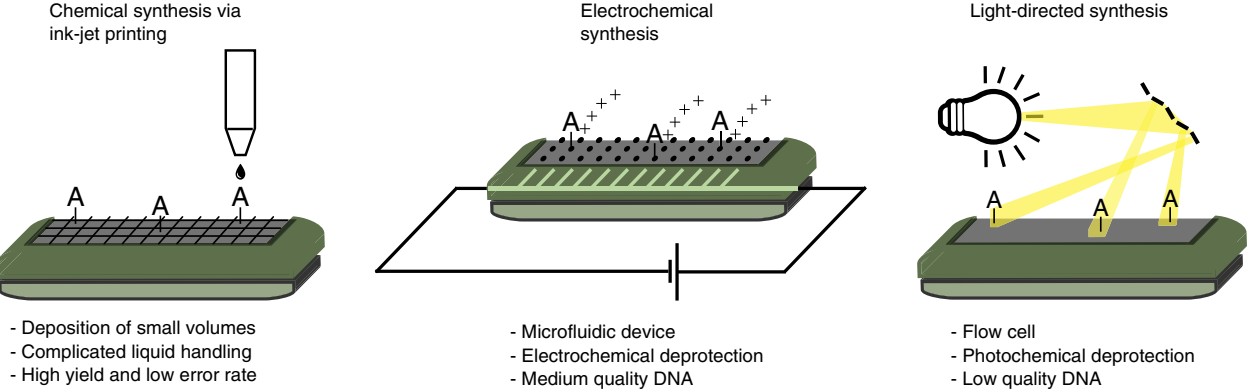

**Fig. 7 Comparison of different DNA synthesis platforms and their characteristic traits.** Printing technology is primarily used by Twist and Agilent. Electrochemical synthesis is employed by Custom Array. This work uses light-directed synthesis.

in Fig. 2d, these technologies can manufacture large pools of DNA sequences with very low-error levels[16].

However, this precision comes at a cost (see Table 1), which may be related to the complexity of the synthesis devices and the reagent excess applied to achieve these oligo purities. For the field of DNA data storage, such high DNA quality is not required, and as shown above, data can be successfully stored and retrieved from DNA that has synthesis error levels in the >5% region. The idea of ligating sequencing adapters to enable data read-out is primarily not restricted to photochemically synthesized DNA. It evolves out of the necessity that constant regions directly synthesized are too erroneous to allow for PCR-based library preparation or instant sequencing. From an automation viewpoint, however, it makes sense to incorporate primer regions into every strand if there is the possibility.

It has to be noted that the cost comparison in Table 1 is not fully fair, as in most approaches DNA was sourced from a

commercial supplier, and the oligo costs will also comprise an overhead. For our work and the work of Lee et al.[3], however, the raw products were priced at the effective rates, which are non-industrial small-scale prices from specialty manufacturers/suppliers with significant room for improvement. While it can be expected that material deposition and electrochemical DNA synthesis methods can be operated at lower precision levels to save cost when preparing oligos for DNA synthesis, photolithographic DNA synthesis has some intrinsic advantages in this mode of operation: the set-up is relatively simple, merely requiring a UV DLP light engine and a straight-forward flow-cell operating on standard glass slides. In contrast, material deposition techniques require the integration of an ink-jet print-head into a flow cell, and this semi-open system has to be run in an anhydrous print chamber[21]. While the electrochemical synthesis method also only requires a standard flow-cell, specifically designed electrode surfaces are required, which, at least in the

**Table 1 Comparison of prior DNA storage work with key parameters.**

| Parameter Synthesis method | Church et al. Deposition (Phosphoramidite) | Goldman et al. Deposition (Phosphoramidite) | Grass et al. Electrochemistry (Phosphoramidite) | Erlich et al. Deposition (Phosphoramidite) | Organick et al. Deposition (Phosphoramidite) | Lee et al. Column (Enzymatic) | This work Light-directed (Phosphoramidite) |
|---|---|---|---|---|---|---|---|
| Input data (MB) | 0.65[1] | 0.75[1] | 0.08[1] | 2.15[1] | 200.2[13] | 1.2 × 10⁻⁵ | 0.1 |
| Net info (bits nt⁻¹) | 0.83[1] | 0.33[1] | 1.14[1] | 1.57[1] | 1.10[13] | 0.46 | 0.94 |
| Info incl. primers (bits nt⁻¹) | 0.60[13] | 0.19[13] | 0.86[13] | 1.18[13] | 0.81[13] | 0.46 | 0.94 |
| Cost (US$ MB⁻¹) | 11,650[2] | 12,400[1] | 31,250[12] | 3260[11] | 5610 | 7010* | 530* |

Note that net information density excludes the primers. Post-synthesis costs, such as library preparation and sequencing are not included in the cost calculations. Detailed information about calculations can be taken from Supplementary Note 5.
*Only consumable costs (at non-industry rates), no overhead or library preparation costs considered.

current mode of operation, are not recycled. Furthermore, photolithographic synthesis does not use highly acidic deblocking reagents and oligonucleotide length is therefore not limited by depurination; this promises long sequence synthesis and minimizes setup time and substrate processing. Estimations for commercial pricing show that in the current development stage, our method is already half as expensive as established market leaders (see Supplementary Discussion).

Comparing current DNA synthesis cost and throughput with other methods of data storage (e.g., magnetic tape), it is evident that significant further advancement is required to be competitive. Latencies of hard-drives and low read-write cycle times will be difficult to achieve. However, advances in DNA-based non-volatile memory combined with logical operations[22] and innovative synthesis methods using e.g., UV patterning and DNA self-assembly[23] help to alleviate this gap. It can be anticipated that improving the number of oligos synthesized in parallel and further optimizations reagent usage will get synthesis costs down to the 1 US\$ MB⁻¹ region rapidly. Further cost improvements can be obtained by scaling the synthesis of the reagents, which are the major cost factors.

Extending on this, the marginal costs of chemical DNA synthesis can be calculated: Assuming that 10,000 copies of every oligo are synthesized simultaneously, a nucleotide reagent cost of 100 US\$ g⁻¹ and overall logical density of 1 bit nt⁻¹, the cost of 1 MB stored is ~1e−8 US\$ at 100% chemical yield. While 100% chemical yield will never be obtainable for such reactions, DNA storage can be cost-competitive with tape storage (0.02 US\$ GB⁻¹) at a chemical yield of 0.1%. This is equivalent to a 1000× reagent excess, which is in line with reaction conditions commonly found in surface chemistry, and shows that further optimized chemical DNA synthesis methods are suited for the application of archival data storage using DNA.

We have shown that significant cost reductions in DNA data storage are possible, when the photolithographic DNA synthesis methodology is combined with an appropriate error correction approach. For this, we developed augmented error correction coding with an advanced read clustering algorithm using locality-sensitive hashing and alignment through majority voting steps to extract good candidates for the error-correcting scheme. Our experimental approach additionally enables DNA synthesis without read adapter sequences, which further cuts required synthesis costs. In comparison to previous approaches in DNA storage, which use high-purity oligos, this work shows that DNA data storage is indeed possible with significantly lower quality oligos, thereby fully profiting from the error correction codes and reconstruction algorithms.

## Methods

**DNA microarray fabrication by photolithography.** The fabrication of DNA microarrays by photolithography follows established protocols but improved through a series of technical adjustments[8,9,24–28]. A first step consists in attaching functional hydroxyl groups on glass microscope slides via silanization using N-(3-triethoxysilylpropyl)-4-hydroxybutyramide (Gelest SIT8189.5). The slides are placed in a drying rack and submerged in a solution (500 ml) of silane (10 g) in EtOH/H$_2$O 95:5 + 1 ml AcOH for 4 h at r.t. After two subsequent washes in EtOH/H$_2$O 95:5 + 1 ml AcOH for 20 min each, the slides are baked and cured overnight in a vacuum oven at 120 °C, then transferred into a desiccator where they are stored until use. Half of the slides are drilled at two precise locations with a 0.9 mm diamond bit with a CNC router (Stepcraft) prior to functionalization, then rinsed with ultrapure water and cleaned in an ultrasonic bath for 30 min. A pair of drilled and undrilled slides are then placed on top of each other, separated by a 50 μm-thick PTFE gasket, and assembled into a synthesis cell which is then attached to the DNA synthesizer (Expedite 8909, PerSeptive Biosystems) controlling the delivery of solvents and reagents to the cell. The cell is then attached to a support located at the focal point of the photolithographic system. The chamber between the drilled microscope slide and the quartz block of the cell is filled with a 1% (w/w) solution of β-carotene in dichloromethane, which acts as a UV absorber, preventing reflection of UV light rays off the quartz block back onto the glass substrates[27]

365 nm UV light is generated by an LED (Nichia NVSU333A)[28] The UV light is spatially homogenized in a mirror tunnel and then imaged onto a Texas Instruments Digital Micromirror Device (DMD), which consists of an array of 1024 × 768 mirrors digitally tiltable into "ON" or "OFF" positions. UV light reflecting on the "ON" mirrors is projected onto the synthesis cell using an Offner relay optical system, and UV light reflecting on the "OFF" mirrors is projected away from the synthesis cell. Illumination of the slides with UV light triggers the removal of the photosensitive benzoyl-2-(2-nitrophenyl)-propoxycarbonyl (Bz-NPPOC) protecting group on the 5′ hydroxyl of DNA phosphoramidites (Orgentis) at the locations corresponding to the pattern of "ON" mirrors. A computer controls the timeframe of UV illumination and communicates the correct pattern of ON and OFF mirrors to the DMD. Light exposure lasts 35 s at an irradiance of ~85 mW cm$^{-2}$, yielding a radiant energy density of 3 J cm$^{-2}$. During UV deprotection, the synthesis cell is filled with a solution of 1% (w/w) imidazole in DMSO to help mediate the removal of the photosensitive group. Only the illuminated positions ("features"), with a now free 5′-OH group, can react with the incoming phosphoramidite for the subsequent coupling step, activated with 0.25 M dicyanoimidazole in acetonitrile (ACN). After coupling (15 s), the slides are rinsed with ACN and the phosphite triester groups are oxidized into phosphotriesters using a mixture of $I_2$ in pyridine/$H_2O$/THF. DNA phosphoramidites are base-protected with *tert*-butylphenoxylacetyl (tac) for dA and dG and acetyl for dC.

**Library design, synthesis, and recovery**. Prior to synthesis, a list of all sequences is transformed into an ordered series of images ("masks"), a pattern of ON and OFF mirrors, using a MATLAB (MathWorks) program designed in-house. The order of masks corresponds to the order of phosphoramidite couplings (or "cycles"), totaling 187 for the synthesis of >16,000 unique sequences. Each feature was 8 × 6 micromirrors large in size. For the synthesis of the corresponding library by microarray photolithography, a $dT_5$ linker is first synthesized everywhere on the silanized substrates, followed by the coupling of a dT phosphoramidite carrying a base-sensitive succinyl moiety at the 3′ position (ChemGenes Corporation, coupling time 2 × 120 s)[29]. This step yields a 3′ T at the end of every sequence. After the 187 coupling cycles mentioned above, a final UV light exposure at the end of the synthesis removes the terminal 5′-Bz-NPPOC groups on all oligonucleotides. After synthesis, the slides are transferred into a 1:1 solution of dry ethylenediamine/toluene (20 ml each) and gently agitated on a shaker in order to remove the protecting groups and to cleave the succinyl ester. After 2 h at r.t., the slides are washed in dry ACN (2 × 20 ml). A small (100 μl) volume of ultrapure water is then carefully applied onto the synthesis area, mixed back and forth and recovered into a 1.5 ml microcentrifuge tube. The procedure is repeated with another 100 μl of water. The aqueous solution of cleaved, deprotected DNA is then evaporated to dryness, rediluted in 10 μl of water and quantified using 260 nm absorption on a NanoVue spectrophotometer (GE Healthcare Life Sciences). This chip eluate was then desalted using a ZipTip $C_{18}$ pipette tip (Millipore). Briefly, the tip is first wetted with 3 × 10 μl ACN/$H_2O$ 1:1, equilibrated with 3 × 10 μl 0.1 M triethylammonium acetate (TEAA) buffer and loaded with the DNA sample by pipetting back and forth 10 times. The ZipTip is then washed with 3 × 10 μl TEAA and 3 × 10 μl $H_2O$. Finally, DNA is eluted from the tip with 10 μl ACN/$H_2O$ 1:1, quantified (49 pmol of DNA isolated from both substrates) and dried down.

**Adapter ligation and library preparation**. In order to render the files readable, adapters for Illumina sequencing (P5 and P7 TruSeq LT Adapter with Indeces I14, I16 and I18, see Supplementary Table 3) were ligated using Accel-NGS 1S Plus DNA Library Kit[14] (Swift Biosciences). In the first step, the truncated adapter was ligated to the 3′ ends of the sequences by combining DNA (15 μL, 24 ng) with the pre-mixed adaptase reaction mix (25 μL) consisting of reagents and enzymes in Tris-EDTA (TE) buffer. The solution was heated to 37 °C for 15 min and subsequently 95 °C for 2 min in a thermocycler. The incorporation of the truncated adapter was conducted by extension of primers. Therefore, the pre-mixed extension reaction mix (47 μL) was added to the previous reaction mixture and then heated to 98 °C for 30 s, 63 °C for 1 min and 68 °C for 5 min. The resulting solution was purified using magnetic beads (157 μL, ratio: 1.8, CleanNA CleanNGS), washed twice with a 1:5 mixture of deionized water (18.20 MΩ·s, Thermo Scientific Micropure UV) and ethanol (500 μL, VWR Chemicals) and eluted in TE buffer (20 μL). The ligation of the truncated adapter to the 5′ end of the strands was achieved with the addition of the pre-mixed ligation reaction mix and subsequent heating to 25 °C for 15 min. The solution was then purified with magnetic beads (64 μL, ratio: 1.6), washed and eluted in TE buffer (20 μL). The final step included polymerase chain reaction (PCR) that incorporates the full length adapters with a single index suitable for Illumina next-generation sequencing (NGS). The pre-mixed indexing PCR reaction mixture was added and the solution was heated to 98 °C for 2 min. Cycling parameters were 98 °C for 10 s, 60 °C for 30 s and 68 °C for 1 min, 16 cycles. The Library sequences have a length of 181 bp after full adapter ligation (see Supplementary Fig. 2d). The three samples were pooled 1:3:8 (File1:File2:File3) and sent for sequencing on an Illumina NextSeq Sequencer (Microsynth AG, CH).

**Synthesis cost calculation**. In the calculation of the price of one synthesis, every chemical, solvent and consumable was taken into account. We have determined the consumption of each component in a synthesis run and calculated the total expense of one component by means of the per unit price. The detailed rundown of costs is provided in Supplementary Table 2 in the Supplementary Information. The calculated price of 52.54 US$ per synthesis does not include any overhead cost as mentioned before.

**Statistical analysis**. No statistical analysis was performed in this work. For Monte-Carlo simulations, 16,383 sequences were used.

**Reporting summary**. Further information on research design is available in the Nature Research Reporting Summary linked to this article.

## Data availability
Raw sequencing data can be retrieved under 10.6084/m9.figshare.c.5128901.v1[15]. Due to its size, file 3 can be made available by the authors upon request. All other relevant data are available upon reasonable request. Source data are provided with this paper.

## Code availability
Code can be downloaded from https://github.com/MLI-lab/noisy_dna_data_storage (DOI: 10.5281/zenodo.4044459)[30].

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

## Acknowledgements
The Austrian Science Fund (FWF P27275 and P30596) is gratefully acknowledged. Rice University, Department of Electrical Engineering, while most of the computational research reported here was performed.

## Author contributions
The research was conceived by R.N.G. and R.H. with inputs from W.J.S. and M.M.S.; DNA synthesis was conducted by J.L. and library preparation and sequencing by P.L.A.; R.H. and M.Z.D. performed encoding and decoding as well as error analysis; P.L.A. conducted Monte-Carlo simulations. The paper was written by P.L.A., R.N.G., and R.H. with contributions from all other authors.

## Competing interests
The authors declare no competing interests.
