## [Peer Review File · Nature Communications]

Reviewers' Comments:

Reviewer #1:

Remarks to the Author:

Antkowiak et al. present a DNA storage work that relies on synthesis of the DNA molecules in an academic setting using photolithography. Unlike other architectures (e.g. Twist), photolithography has no large mechanical parts and offers a dramatically smaller feature size, which can substantially reduce the costs. In addition, photolithography does not use acid washes that are common in regular phosphoramidite synthesis and therefore is not subject to de-purination build-up that effectively restrict the lengths of the synthesized molecules. On the other hand, photolithography is more error-prone than regular phosphoramidite synthesis and therefore needs a robust error correcting code.

The manuscript is of an interest, creative, and has several merits such as analyzing the error rates, developing a trace reconstruction procedure to correct the reads, and using Swift Biosciences product to alleviate the synthesis of PCR primers (very smart!). I think it will be a highly valuable contribution to the field.

Major points

Point 1:

The main caveat of the manuscript is that it did not demonstrate that photolithography is a worthy alternative from a cost reduction perspective, which is the main point. Let me explain:

A. Table 1 shows that the synthesis costs of their setting were \$530/Mb compared to \$3260/Mb for a library synthesized by Twist in 2016, which is the lowest among all other alternatives. However, this comparison is not fair (as the authors themselves noted) since they compare costs of goods versus the final price of a product. This is an apple-to-orange comparison. The final price typically takes into account capital expenses (the cost of building the machine) amortized over the life time of the machine, the operating costs, personnel, marketing expenses, sales tax, and some profit margin.

B. On top of that, Twist now offers a much better deal of DNA synthesis which is around \$1500/Mbyte (about half the price of 2016), so the price is much lower these days.

C. Finally, some of the cost reductions that the authors achieved is from the super-smart idea of eliminating the adaptors. However, this clever idea can be used in any type of DNA synthesis and not unique to photolithography.

So all in all, if we factor points (a)-(c), it is quite likely that the difference between the costs of goods of their current experiments and the offering by other existing technologies is not dramatic. Estimating the true costs of other types of DNA synthesis technologies is hard and I do not want to discourage the authors or request a task that cannot be accomplished.

The manuscript will be much stronger and convincing if the authors can demonstrate a difference in costs that is substantially bigger (e.g. \$25/Mbyte versus Twist final price of \$1500/Mb).

In the current manuscript, the authors did not fully utilize the DMD feature size and instead created each oligo using 6x8 pixels. This is highly wasteful design that means that the costs can go down by 48-fold. In addition, the experiment generated only ~16,000 oligos, which is quite small and does not stress test the scalability of the decoding mechanism.

Therefore, I strongly recommend that the authors will conduct another DNA storage experiment but this time utilizing approximately 50% of the features (e.g. checkerboard pattern) and the same information content instead of 2% of the features as in the current experiment. Such an experiment should cost only a few hundred dollars in reagents since it will utilize the same setting.

Point 2:

In addition, as this is a methods manuscript, the authors must make their code publicly available using a suitable license according to Nature Communication guidelines and also the sequencing data. The manuscript states the software will be delivered on request but my review package included software (but no license was specified!). The best solution is to simply upload the software to GitHub and include an appropriate open source license (e.g. MIT or GPLv3).

Minor points:

1. How does the Swift product works? The authors give a link to the product but it is unclear what type of biochemical modifications occur that enable single strand amplification. Can you please elaborate since this is one of the strongest points of the manuscript?
2. Why did you use Bz-NPPOC and not SPh-NPPOCC as photolabile group? The Somoza lab has demonstrated that this reagent is superior, so a short discussion to justify your setting would make the manuscript more complete.
3. Please add a full description of the cost model in the Method section (i.e. how did you get to \$500/Mb? How much each reagent cost?)
4. Figure 1 is great. Why did you select this non-standard configuration of 6bits as your "atomic unit"? why not 8bits?
5. I also suggest to analyze the error rate as a function of the x,y coordinate of the oligo on the DMD. This will allow to see whether the center of the DMD produces more accurate results versus the sides.
6. A short discussion on the scalability of your trace reconstruction for a larger number of oligos will be highly useful.

Reviewer #2:

Remarks to the Author:

Generally, I very much support new work regarding potentially novel applications of synthetic DNA to non-biological uses like memory and data storage. However, that being said most work does not even begin to approach a level applicability that is necessary for practical applications--- where it would be competitive with present micro/nanoelectronic forms of logic and memory devices.

As research report, I thought the paper presented some new approaches that further advance the area. However, I had two issues. First, while talking about cost reduction for DNA synthesis has some validity, it is only one small part of what would be necessary for actually producing a viable DNA memory/data storage technology. The fact that read-out requires producing NGS libraries and subsequent sequencing is "major" limitation. In order to compete with present micro/nanoelectronic logic and memory devices, any new technology like DNA memory has to have a major advantage that the present technology does not meet. Certainly, read out of DNA information is that limitation. Secondly, The authors left out some other important references of recent DNA memory related publications; which are: Song Y, Kim S, Heller MJ and Huang X, "DNA multi-bit non-volatile memory and bit-shifting operations using addressable electrode arrays and electric field-induced hybridization", Nature Communications (2018) 9:281, DOI: 10.1038/s41467-017-02705-8; and Song Y, Takahashi T, Kim S, Heaney YC, Warner J, Chen S and Heller MJ, A Programmable DNA Double-Write Material: Synergy of Photolithography and Self-Assembly Nanofabrication, DOI: 10.1021/acsami.6b11361, ACS Appl. Mater. Interfaces 2017, 9, 22–28. These approaches involve the use of electronic microarray fabrication and uv photolithography. However, they are fundamentally different from the three approaches the authors presented in Figure 5 of the manuscript. More importantly, the new references begin to address the problem of information read-out.

Overall, I do like the paper as it is good work and it does advance the area. However, I would only

ask that the authors might add some "realistic" comments about the other problems that need to be solved before DNA can be used for these non-biological applications.

Response to Reviewers

Comments in *blue*, Replies in black, Actions in **bold**

Reviewer #1:

Remarks to the Author:

Antkowiak et al. present a DNA storage work that relies on synthesis of the DNA molecules in an academic setting using photolithography. Unlike other architectures (e.g. Twist), photolithography has no large mechanical parts and offers a dramatically smaller feature size, which can substantially reduce the costs. In addition, photolithography does not use acid washes that are common in regular phosphoramidite synthesis and therefore is not subject to de-purination build-up that effectively restrict the lengths of the synthesized molecules. On the other hand, photolithography is more error-prone than regular phosphoramidite synthesis and therefore needs a robust error correcting code.

The manuscript is of an interest, creative, and has several merits such as analyzing the error rates, developing a trace reconstruction procedure to correct the reads, and using Swift Biosciences product to alleviate the synthesis of PCR primers (very smart!). I think it will be a highly valuable contribution to the field.

We would like to thank the referee for the positive reaction on our work. In the revised manuscript, we have added a stronger experimental foundation and a reliable scaling model that provides insight in the practical possibilities of the described synthesis technology and data storage channel. We included calculations on the needed data redundancy that are based on experiments with larger files. Furthermore, we have added a comprehensive analysis on library preparation and more insight in synthesis quality. We hope the thorough and extensive improvements will increase the value of this work.

Specific comments:

Major points:

- 1) *The main caveat of the manuscript is that it did not demonstrate that photolithography is a worthy alternative from a cost reduction perspective, which is the main point. Let me explain:*
 - A. *Table 1 shows that the synthesis costs of their setting were \$530/Mb compared to \$3260/Mb for a library synthesized by Twist in 2016, which is the lowest among all other alternatives. However, this comparison is not fair (as the authors themselves noted) since they compare costs of goods versus the final price of a product. This is an apple-to-orange comparison. The final price typically takes into account capital expenses (the cost of building the machine) amortized over the life time of the machine, the operating costs, personnel, marketing expenses,*

sales tax, and some profit margin.

We thank the reviewer for this valuable comment. The comparison is, in fact, not fair as already mentioned in the main text.

In order to make the costs more comparable, we estimated a consumer price based on our setup which is still factor 2 less expensive than market competitors at the moment. A detailed run-down and comparison with leading synthesis companies can be found in the Supplementary Discussion:

“For a better comparison of costs concerning our method and prices of competitors already established on the market, we performed a rough cost estimation. We chose list prices that do not include academic discounts for all mentioned consumer prices. The data encoded was calculated from the number of features times the net info which was set to 1 bit/nt for all methods. For CustomArray we chose the biggest chip with 92'918 features and a net length of 130 nt. Twist Biosciences' chip with 600'000 features and a net length of 210 has the best trade-off between price and amount of encoded data and in the case of Agilent, a chip with 244'000 features and a net length of 190 nt was chosen. The net length takes into account 20 nt on 5' and 3' end, needed for sequencing primer ligation. Full adapter ligation as employed in this work is primarily not restricted to photo-directed synthesis. However, for automation purposes, it remains questionable if there is a cost advantage by using it for established synthesis techniques. Here, commonly applied working schemes are used to compare the state-of-the-art. While we cannot guess the machine cost, personnel cost, marketing cost and other costs for the already existing suppliers, we have access to the final sales price, and can compare this with a sales price using a cost model for the maskless DNA synthesis presented here.

Table S3: Cost estimations for industrial-scale photo-directed synthesis machine in comparison with established suppliers. (See Supp. Info for table)

Equipment cost for our synthesis machine were roughly 100.000 US\$. Assuming the annual capital expenses (CAPEX) are about four times as much¹⁷, which includes the time value of money and a conservative depreciation over 10 years. A machine that is running 300 syntheses every year therefore costs approximately 1333 US\$/synthesis. A detailed cost rundown for the chemicals needed in a synthesis can be found in Table S2. For the personnel, 4 h/synthesis including quality control at 50 US\$ h⁻¹ and a conservative 100% overhead cost for electricity, buildings, facilities, etc, were estimated. Electricity costs for the DNA synthesis are included here. This cost was copied 1:1 for marketing expenses. A universal VAT (value added tax) of 10% and a 10% profit margin were included. For the experiment with a data load of 1.3 MB which can still be scaled to having a net density of ~0.3 bits/nt on the same machine, a price of 2036 US\$/MB can be calculated. This shows that even with conservative cost calculations the synthesis method employed in this work is already superior by a factor of ~2. Taking into consideration the development status and great scaling potential, considerable improvements can be achieved.”

We want to emphasize again that the intrinsic advantages of our simple setup, no need of acidic deblocking agents combined with good scalability and error correcting codes, this method is a real alternative to established techniques.

B. On top of that, Twist now offers a much better deal of DNA synthesis which is around \$1500/Mbyte (about half the price of 2016), so the price is much lower these days.

Twist still offers the same list price as in 2018. They now offer 300 nt long oligos for the price of 114'422 USD which adds about 15 % of cost compared to the 3260 USD described in the manuscript in Table 1. **In order to compare market prices, that do not include academic discounts, we estimated the cost of our synthesis method which can be found in the Supplementary Discussion.** We have indicated in the manuscript that the numbers presented in Table 1 are not fully fair, but show the potential of this technique.

C. Finally, some of the cost reductions that the authors achieved is from the super-smart idea of eliminating the adaptors. However, this clever idea can be used in any type of DNA synthesis and not unique to photolithography.

We thank the referee for this important note. **This issue has been addressed in the main text:**

“The idea of ligating sequencing adapters to enable data read-out is primarily not restricted to photochemically synthesized DNA. It evolves out of the necessity that constant regions directly synthesized are too erroneous to allow for PCR-based library preparation or instant sequencing. From an automation viewpoint, however, it makes sense to incorporate primer regions into every strand if there is the possibility.”

It is difficult to estimate the influence on the cost structure of conventionally synthesized DNA as it belongs both to the apparent synthesis costs and the expenses to read the data.

So all in all, if we factor points (a)-(c), it is quite likely that the difference between the costs of goods of their current experiments and the offering by other existing technologies is not dramatic. Estimating the true costs of other types of DNA synthesis technologies is hard and I do not want to discourage the authors or request a task that cannot be accomplished. The manuscript will be much stronger and convincing if the authors can demonstrate a difference in costs that is substantially bigger (e.g. \$25/Mbyte versus Twist final price of \$1500/Mb). In the current manuscript, the authors did not fully utilize the DMD feature size and instead created each oligo using 6x8 pixels. This is highly wasteful design that means that the costs can go down by 48-fold. In addition, the experiment generated only ~16,000 oligos, which is quite small and does not stress test the scalability of the decoding mechanism.

Therefore, I strongly recommend that the authors will conduct another DNA storage experiment but this time utilizing approximately 50% of the features (e.g. checkerboard pattern) and the same information content instead of 2% of the features as in the current experiment. Such an experiment should cost only a few hundred dollars in reagents since it will utilize the same setting.

We agree with the referee in this point and therefore contemplated the synthesis of larger files that would scale down the synthesis cost (with the same coding scheme and the same cost of raw materials) to 160 USD/MB and 40 USD/MB,

respectively. **We conducted two more experiments with file sizes of 323 kB and 1.3 MB respectively.** Due to the error prone synthesis, which specifically produces more insertion errors with a larger amount of digital micromirrors used per area, this requires the coding scheme adapted to a higher redundancy. For this reason, **we included a discussion in the subsection ‘Scaling to larger files’ that considers the feasibility of synthesizing and sequencing a more diverse pool of DNA with this technology, and gives new data on error rates for larger synthesis experiments, which allow the calculation of the required redundancies:**

“Scaling to larger files. So far, we focused on a file of about 100 kB stored on 16383 sequences. In order to understand how our method scales to larger amounts of data, we performed two additional experiments where we stored about 323 kB and 1.3 MB on 49149 and on 196596 sequences respectively, with exactly the same coding scheme and with the same parameters (in particular the same amount of redundancy) as described before. We refer to those files as file 2 and file 3, respectively, and to the original 100 kB file as file 1. For file 2 and 3 we obtained 100,706,815 and 195,619,515 many reads, respectively. Compared to file 1, the error probabilities in the sequences is higher, specifically, we found the insertion probabilities to increase by about 2% and 4% for files 2 and 3 relative to file 1, whereas the substitution and deletion errors remained relatively constant (see Fig. 6, and compare to Fig. 2 panel c). Note that we expect the error probability to be higher since when synthesizing a larger amount of sequences, the spots on the array become smaller, and thus we expect lower-quality synthesis.

Due to the higher amount of errors, we were not able to recover the information, but would have been able by using more redundancy in our coding scheme. In more detail, after clustering the reads of file 2 and file 3, 35% and 31% of the original sequences appear at least once without error, as opposed to 83% for file 1. By increasing the redundancy of the outer code by a factor of about 3, we would be able to recover the data perfectly from those measurements. In this case, costs would be further minimized to 160 MB/US\$ for file 2 and 40 MB/US\$ for file 3.”

Fig. 6: Read analysis for files of larger size. a,b Read length distribution for file2 and file3, respectively. **c,d** Error probability of reads cut after 60 nt in 5'-3' direction for file2 and file3, respectively.

In addition, as this is a methods manuscript, the authors must make their code publicly available using a suitable license according to Nature Communication guidelines and also the sequencing data. The manuscript states the software will be delivered on request but my review package included software (but no license was specified!). The best solution is to simply upload the software to GitHub and include an appropriate open source license (e.g. MIT or GPLv3).

As requested by nature submission guidelines and also upon the editor's request we provide the code on GitHub under an Apache-2.0 license:

https://github.com/MLI-lab/noisy_dna_data_storage

Minor points:

- 1) *How does the Swift product works? The authors give a link to the product but it is unclear what type of biochemical modifications occur that enable single strand amplification. Can you please elaborate since this is one of the strongest points of the manuscript?*

The referee raises an important point with the lack of insight given concerning the library preparation. **We included a comprehensive paragraph in the manuscript elaborating on the principle of single strand ligation of sequencing adaptors:**

“The underlying principle of the library preparation out of a random ssDNA pool without a constant region is the ligation of template-mediated addition of deoxyribonucleoside monophosphates (dNMPs) including an attenuator. The latter will control the number of attached dNMPs and additionally comprises part of an NGS adaptor double-stranded sequence. The addition of bases to the 5' end of the ssDNA is directed by terminal deoxynucleotidyl transferase (TdT) and concomitant ligation of both ends. Similarly, to the 3' end of the oligonucleotide, a DNA polymerase adds bases complementary to a homopolymeric attenuator-adaptor construct that is ligated in the same step. In this way, the treated ssDNA can undergo the full adaptor ligation in a PCR that also incorporates read indices.”

Further, we conducted extensive experimental research to gain more knowledge on our specific use of the Swift Biosciences product. The experiments and data are laid out Supplementary Note 2:

“To further shed light on the functionality of the commercial kit used to prepare sequencing libraries out of ssDNA, gel electrophoresis pictures in agarose gel of different stages within the preparation were recorded. Due to abundance and better visibility in gel electrophoresis, model DNA of two different lengths were employed and the maximum input amount of 250 ng was taken. For this purpose, a 60mer (Microsynth AG, CH) where each base position is randomly distributed, representing a very similar DNA molecule was synthesized conventionally. Another 158 bp long dsDNA random library was used as a reference for the comparison of longer starting templates and double stranded inputs. The two DNA samples

serve as dummies to investigate how enzymes attach read adapters and to test overall kit robustness and integrity. Fig. S1 shows the process employed in the preparation of a library.

Fig. S1: Workflow of the Accel-NGS 1S Plus DNA Library Kit (Swift Biosciences) adapted from the manual. DNA fragments first undergo 3' ligation of truncated adapter (1) and duplexing (2), followed by 5' ligation of truncated adapter (3) and finally PCR (4) that concatenates the full-length adapters.

First, the template ssDNA fragments are heated for 2 min to 95 °C and immediately put on ice to ensure only single strands partaking in the first reaction. A truncated adapter is ligated to the 3' prime end of the template. The reaction mixture essentially consists of a template independent polymerase in the form of terminal deoxynucleotidyl transferase (TdT) which is inhibited in their activity by mainly two factors: a suitable reaction environment controlled by ion concentrations without inhibiting the activity of involved enzymes and an attenuator-adapter complex. TdT adds single nucleotides to the 3' end of the template DNA which is directed by the composition of the reaction solution providing primarily deoxycytidine and deoxythymidine monophosphate (dCMP and dTMP). The attenuator part of the inhibiting complex is complementary to the CT tail growing in this manner and additionally stops uncontrolled growth of the tail with a blocking group that prevents more bases from being bound to the template. It further comprises a section adjacent to the attenuator sequence that is complementary to a part of an NGS adapter sequence (Truncated adapter P7, see Fig. S1) which is also present in the reaction mixture. Another enzyme concatenates the 3' CT tail end and the truncated adapter sequence in place. Enzymes and chemicals for the second step that is rendering the template-adapter intermediate double stranded, is directly applied to the reaction mixture without purification. The bulk of the reaction takes place at 37 °C (15 min) with a 2 min denaturation step at 95 °C. Fig. S2b shows the length of the templates after these first two steps. The templates are now around 100 bp for the 60mer and around 200 bp for the 158mer, revealing a truncated adapter length of ca. 40 bp which is about 2/3 of the full-length adapter. Similarly, the P5 truncated adapter is ligated to the 3' end of the template in the third step allowing the use of PCR primers completing adapter addition and significantly increase the copy number of the template strands (Step 4). Template length after Step 3 is ~130 bp for the 60mer and ~ 230 bp for the 158mer, showing a P5 truncated adapter length of ca. 30 bp (approximately half of the full-length adapter). Especially this step is not straight-forward to see in the gel pictures. The reason for this is probably that the attached adapter part is small compared to the rest of the molecule and single-stranded which likely doesn't change elution properties to an extent that can be easily detected. The ready-to-sequence product after PCR has a target length of 181 bp for the

60mer and 279 bp for the 158mer. This can be confirmed by gel electrophoresis (see Fig. S2d).

Fig. S2: Agarose gel (2%, Invitrogen E-Gel EX 2%) images. The left lane always shows the random 60mer oligo, the middle lane shows the 158mer double-stranded library and the right lane shows a 50 bp DNA ladder (Thermo Scientific GeneRuler) as a reference. (a) Original input (b) Fragments after step 2 of the library preparation. (c) Fragments after step 3 of the library preparation. (d) Final product after PCR, ready-to-sequence. The images are compiled from several different gel electrophoresis runs. Original images can be found in the electronic supplementary information.

- 2) Why did you use Bz-NPPOC and not SPh-NPPOC as photolabile group? The Somoza lab has demonstrated that this reagent is superior, so a short discussion to justify your setting would make the manuscript more complete.

We chose to perform microarray photolithography with Bz-NPPOC rather than SPh-NPPOC phosphoramidites to keep with our cost-effective approach of DNA library synthesis. Indeed, Bz-NPPOC monomers are commercially available (Orgentis Chemicals) and cost ~€70/g, whereas SPh-NPPOC need to be custom-made and are significantly more expensive, at around ~€360/g. Given that amidite cost accounts for ~30% of total chemical costs, using SPh-NPPOC amidites would have made the library synthesis considerably more expensive.

- 3) Please add a full description of the cost model in the Method section (i.e. how did you get to \$500/Mb? How much each reagent cost?)

We thank the referee for this valuable point that is now **described in the section ‘Synthesis Cost calculation’ in the main text. We also added a complete list of chemicals and prices that form the basis of our synthesis cost calculations** (Supplementary Table S2):

“Synthesis Cost Calculation In the calculation of the price of one synthesis, every chemical, solvent and consumable was taken into account. We have determined the consumption of each component in a synthesis run and calculated the total expense of one

component by means of the per unit price. The detailed rundown of costs is provided in Table S2 in the Supplementary Information. The calculated price of 52.54 US\$ per synthesis does not include any overhead cost as mentioned before.”

Table S2: Detailed description of consumption and prices used for one synthesis. (see Supp. Info)

4) Figure 1 is great. Why did you select this non-standard configuration of 6bits as your “atomic unit”? why not 8bits?

The reason we used 6 bits is that the underlying inner code has symbols in $\{0, \dots, 2^{6-1}\}$ i.e., it has symbols that are 6 bit long. We could of course use a code with symbols that are 8 bits long, but 6 gave a better trade-off with regards to the number of errors that can be corrected relative to the amount of redundancy added. It should be noted that we can simply map input data in byte (i.e. organized in blocks of 8 bit) to data organized in blocks of 6 bit, without occurring any loss or other inefficiency.

5) I also suggest to analyze the error rate as a function of the x,y coordinate of the oligo on the DMD. This will allow to see whether the center of the DMD produces more accurate results versus the sides.

We thank the referee for this excellent idea on error analysis linked to the synthesis device. **We therefore analyzed the edit distance of reads to the original sequence mapped to the synthesis position on the photolithographic chip and the number of reads after clustering for each position.** Figures and interpretation can be observed in the main body of this work:

“To better understand how errors are introduced during synthesis, we analyzed the edit distance and number of reads corresponding to the position of the micromirror on the DMD. Fig. 4 shows the spatial distribution of these characteristic numbers.

Fig. 4: Spatial error analysis of photo-directed synthesis machine. a: Edit distance of reads to the original sequence mapped to the synthesis position on the photolithographic chip. **b:** Number of reads after clustering for each position. 128x128 = 16384-1 sequences were synthesized, resulting in a file size of 99.1 kB.

It is apparent that there is significantly higher production of DNA in a concentric manner, leaving the edges and especially the corners with a low read output (see Fig. 4b). This can be explained by optical losses through UV LED intensity and inefficiencies in the light homogenizer, as well as uneven covering of the UV absorbing β -carotene. Less light arriving on the synthesis surface results in a lower coupling efficiency. On the other hand, the readable sequences have a more homogeneous error distribution over the entire synthesis surface. Generally, reflection and scattering effects decrease synthesis integrity¹⁸ due to the increased mutation probability. A grid shaped pattern could stem from inefficient light homogenizing as well. Another explanation are non-random features during synthesis of index position that have a similar grid pattern. Neighboring features that bond the same nucleotide in one step and are unintentionally photo-exposed would show a slightly reduced error rate.

It should be noted that there is a trade-off between deletion error levels and overall DNA integrity which could be improved by tuning the synthesis procedure. Longer light exposure making deletion errors less likely could in turn lead to photoinduced damage to the DNA.”

6) A short discussion on the scalability of your trace reconstruction for a larger number of oligos will be highly useful.

A discussion on the scalability for larger DNA pools can be found in the section ‘Scaling to larger files’.

Reviewer #2:

Remarks to the Author:

Generally, I very much support new work regarding potentially novel applications of synthetic DNA to non-biological uses like memory and data storage.

We are pleased to find that the reviewer described our work as an impulse for novel synthetic applications.

However, that being said most work does not even begin to approach a level applicability that is necessary for practical applications--- where it would be competitive with present micro/nanoelectronic forms of logic and memory devices. As research report, I thought the paper presented some new approaches that further advance the area. However, I had two issues.

Specific comments:

- 1) *First, while talking about cost reduction for DNA synthesis has some validity, it is only one small part of what would be necessary for actually producing a viable DNA memory/data storage technology. The fact that read-out requires producing NGS libraries and subsequent sequencing is "major" limitation. In order to compete with present micro/nanoelectronic logic and memory devices, any new technology like*

DNA memory has to have a major advantage that the present technology does not meet.

While we appreciate the reviewer's comment about the relevance of our work in the DNA data storage field, this paper does not make a claim to provide a full scale storage system which is comparable to current digital storage technologies. **We included the following comments in the main text, discussing the big gap between a competitive storage system in DNA and current working technologies:**

“Comparing current DNA synthesis cost and throughput with other methods of data storage (e.g. magnetic tape), it is evident that significant further advancement is required to be competitive. Latencies of hard-drives and low read-write cycle times will be difficult to achieve. However, advances in DNA-based non-volatile memory combined with logical operations¹ and innovative synthesis methods using e.g. UV patterning and DNA self-assembly² help to alleviate this gap.”

We are referring to 1: Song et al. (DOI: [10.1038/s41467-017-02705-8](https://doi.org/10.1038/s41467-017-02705-8)) and 2: Song et al (DOI: [10.1021/acsami.6b11361](https://doi.org/10.1021/acsami.6b11361)).

Certainly, read out of DNA information is that limitation.

From a time constraint point-of-view synthesis and readout are certainly both limiting, especially in comparison with today's mass storage media. Economically, this is debatable: Writing with our method would cost around 40 US\$/Megabase. According to NIH, the cost for reading is 10⁻² US\$/Megabase (August 2019, <https://www.genome.gov/about-genomics/fact-sheets/DNA-Sequencing-Costs-Data>) which is factor 4000 less expensive than writing. A visual price comparison until the year 2018 can also be found here: http://www.synthesis.cc/synthesis/2016/03/on_dna_and_transistors.

The cost of writing is paid in the beginning whereas the cost of reading is distributed throughout the life-time of the DNA storage. Considering the time value of money, writing therefore also has a much larger impact on the overall cost than read-out.

- 2) The authors left out some other important references of recent DNA memory related publications; which are: Song Y, Kim S, Heller MJ and Huang X, “DNA multi-bit non-volatile memory and bit-shifting operations using addressable electrode arrays and electric field-induced hybridization”, Nature Communications (2018) 9:281, DOI: 10.1038/s41467-017-02705-8; and Song Y, Takahashi T, Kim S, Heaney YC, Warner J, Chen S and Heller MJ, A Programmable DNA Double-Write Material: Synergy of Photolithography and Self-Assembly Nanofabrication, DOI: 10.1021/acsami.6b11361, ACS Appl. Mater. Interfaces 2017, 9, 22–28. These approaches involve the use of electronic microarray fabrication and uv photolithography. However, they are fundamentally different from the three*

approaches the authors presented in Figure 5 of the manuscript. More importantly, the new references begin to address the problem of information read-out.

As already mentioned in the above comment, **we included the important impulses to the field and also discuss this in the manuscript:**

“Latencies of hard-drives and low read-write cycle times will be difficult to achieve. However, advances in DNA-based non-volatile memory combined with logical operations¹ and innovative synthesis methods using e.g. UV patterning and DNA self-assembly² help to alleviate this gap.”

Overall, I do like the paper as it is good work and it does advance the area. However, I would only ask that the authors might add some "realistic" comments about the other problems that need to be solved before DNA can be used for these non-biological applications.

We are delighted by the reviewer's comment on the quality of our work. We hope that the in-depth discussions on error-prone synthesis and needed redundancy and error-correction coding, the comparison with other synthesis technologies currently in use for DNA-based memory and lastly the enormous potential to scale, will help to bring the advances shown here, into a broader perspective.

Reviewers' Comments:

Reviewer #3:

None